# Patterns of Whole Exome Sequencing in Resected Cholangiocarcinoma

**DOI:** 10.3390/cancers13164062

**Published:** 2021-08-12

**Authors:** Lucas W. Thornblade, Paul Wong, Daneng Li, Susanne G. Warner, Sue Chang, Mustafa Raoof, Jonathan Kessler, Arya Amini, James Lin, Vincent Chung, Gagandeep Singh, Yuman Fong, Laleh G. Melstrom

**Affiliations:** 1Department of Surgery, City of Hope National Medical Center, Duarte, CA 91010, USA; lthornblade@coh.org (L.W.T.); paulwong@coh.org (P.W.); suwarner@coh.org (S.G.W.); mraoof@coh.org (M.R.); gsingh@coh.org (G.S.); yfong@coh.org (Y.F.); 2Department of Medical Oncology, City of Hope National Medical Center, Duarte, CA 91010, USA; danli@coh.org (D.L.); vchung@coh.org (V.C.); 3Department of Pathology, City of Hope National Medical Center, Duarte, CA 91010, USA; suchang@coh.org; 4Department of Radiation Oncology, City of Hope National Medical Center, Duarte, CA 91010, USA; jkessler@coh.org (J.K.); aamini@coh.org (A.A.); 5Division of Gastroenterology, City of Hope National Medical Center, Duarte, CA 91010, USA; jalin@coh.org

**Keywords:** cholangiocarcinoma, tumor genomic sequencing, whole exome sequencing

## Abstract

**Simple Summary:**

Cholangiocarcinomas are rare cancers that harbor a significant number of potentially targetable mutations. In this study, we assessed the frequency of genomic profiling for resected cholangiocarcinomas. We found that, over the past decade, a third of patients underwent tumor genomic profiling, among whom 89% harbored a targetable mutation. Mutations were associated with a median of one approved drug. A quarter of eligible sequenced patients were treated with therapy targeting tumor-specific mutations.

**Abstract:**

Background: With minimally effective chemotherapy options, cholangiocarcinoma patients have 5 year survival rate of 10%. Tumor genetic profiling (TGP) can identify mutations susceptible to targeted therapies. We sought to describe the use of TGP and frequency of actionable results in resected cholangiocarcinoma. Methods: A retrospective review of patients undergoing curative intent resection at a comprehensive cancer center (2010–2020). Clinicopathologic and partial or whole exome sequencing data were reviewed. Results: 114 patients (mean age 65 ± 11 years, 45% female) underwent resection of cholangiocarcinoma (46% poorly differentiated, 54% intrahepatic, 36% node positive, 75% margin negative). Additionally, 32% of patients underwent TGP, yielding a mean of 3.1 actionable mutations per patient (range 0–14). Mutations aligned with a median of one drug per patient (range 0–11). Common mutations included *TP53* (33%), *KRAS* (31%), *IDH1/2* (14%), *FGFR* (14%), and *BRAF* (8%). Targeted therapies were administered in only 4% of patients (23% of eligible sequenced patients). After a median 22 months, 23% had recurrence and 29% were deceased. Discussion: TGP for cholangiocarcinoma has increased over the last decade with targeted therapies identified in most sequenced tumors, impacting treatment in a quarter of eligible patients. Precision medicine will play a central role in the future care of cholangiocarcinoma.

## 1. Introduction

While cholangiocarcinoma represents only 3% of gastrointestinal malignancies, its incidence is increasing worldwide [1,2,3]. The overall prognosis for patients with cholangiocarcinoma remains poor with a 5 year survival rate of approximately 8–10% [4]. Due to its insidious growth, most cholangiocarcinomas are typically diagnosed at a late stage, and only 30% are eligible for curative intent resection at diagnosis due to vascular involvement or intraperitoneal, lymphatic, or distant spread [2,3]. While surgery remains the only curative option for these patients, rates of post-surgical recurrence are high (>60%) [2,5]. Therefore, surgery alone is unlikely to be a definitive modality for treatment in the future. As more is understood about the biology of these tumors, identifying effective systemic therapies, including chemotherapy, immunotherapy, and targeted therapies, will play a key role in the treatment of nearly all biliary tract cancers [2].

Despite decades of investigation, three recent trials have demonstrated no overall survival benefit to adjuvant cytotoxic systemic therapies for resected cholangiocarcinoma (BCAT trial, PRODIGE 12-ACCORD 18-UNICANCER-GI trial, BILCAP trial) [6,7,8]. Regardless of the fact only limited data exists to support its effectiveness, capecitabine is now established as the new standard of care for adjuvant therapy following curative intent resection for cholangiocarcinoma [9]. While the combination of gemcitabine and cisplatinum is considered the standard for palliative treatment based upon the ABC-02 and BT22 trials [10,11], the search for more effective first-line agents is ongoing [12]. Immunotherapy is currently being explored in biliary tract cancers, particularly in tumors with a high tumor mutational burden [13]; however, mismatch repair deficiency and microsatellite instability in cholangiocarcinoma are uncommon (<5%) [14].

There is a desperate need for new targets for treatment of cholangiocarcinoma [15]. In light of this need, it is noteworthy that cholangiocarcinoma is a mutation-rich cancer with a relatively high frequency of mutations [16]. Some have estimated that two-thirds of intrahepatic cholangiocarcinomas harbor actionable mutations [17]. To date, there have been a number of investigations into targeted therapies in the general population of patients with biliary cancer (HER/ERBB, HGF/c-MET, mTOR, etc.) but most have been largely unsuccessful [18]. Due to the failure of trials of targeted therapies in biologically unselected patients, there is a growing interest in testing such therapies on patients with specific mutations [12]. A precision medicine-based strategy for cholangiocarcinoma is supported by data from the MOSCATO-01 trial, which identified druggable mutations in 68% of patients with advanced cholangiocarcinoma (primarily intrahepatic) and demonstrated a median overall survival benefit of 17 months compared with 5 months among patients who were treated with therapy not matched to any mutation (*p* = 0.008) [19].

Mutational analysis has been widely available to clinicians for more than a decade as a means of identifying mutations and to connect patients to clinical trials related to their personal mutations [20]. It is not known how clinicians have used these genomic sequence analyses in patients with bile duct cancers or how commonly patients are directed towards targeted therapies matching identified mutations. We hypothesized that the application of genetic sequence analysis has increased over recent years, but that use of targeted therapies in cholangiocarcinoma remains relatively uncommon. We sought to understand patterns of use and mutations among patients treated with surgical intent at a tertiary cancer center.

## 2. Materials and Methods

We performed a retrospective case series of all patients treated for cholangiocarcinoma via curative intent resection at a tertiary cancer center between 2010 and 2020. Patients were included if they had pathologically confirmed intrahepatic, hilar, or extrahepatic ductal cholangiocarcinoma. Ampullary carcinomas, gallbladder carcinomas, and hepatocellular carcinomas were excluded. Patients were excluded if they never underwent any surgical resection or if surgery was performed with palliative intention only. The cases were reviewed by trained abstractors for inclusion and charts sampled for patient demographics, cholangiocarcinoma risk factors, and tumor features and staging. The patient charts were all sampled for the use of TGP. Genomic sequencing data was collected, including number of genes sequenced, tumor mutational burden (TMB), the number of actionable mutations identified, and the number of drugs associated with those identified mutations. For those genomic sequencing tests that reported such detail, we recorded the number of clinical trials associated with identified mutations. For each identified mutation, we documented the gene and specific codon mutation.

Patient treatment details were collected, including the use of systemic chemotherapy, radiation therapy, and immunotherapies. Surgical resection details included type of resection and margin status. Lastly, any targeted therapies used in treatment—in particular, therapies associated with identified mutations—were documented. We estimated the rate of targeted therapy use among eligible patients, i.e., patients with recurrent or persistent (R1+) disease following surgery who had targetable mutations identified upon sequencing. Patient charts were reviewed for evidence of cancer recurrence and/or death at date of last follow-up. The differences between patients who did and did not undergo genomic sequence testing were reported by a two-sided Student’s *t*-test (continuous variables) and Chi-squared tests (categorical variables). Log-rank tests were used to estimate differences in overall survival. This study was approved by the Institutional Review Board at City of Hope National Medical Center. All statistical analysis was performed using commercially available software (Stata v16.1, StataCorp LLC, College Station, TX, USA).

## 3. Results

A total of 114 patients were treated with curative intent resection for cholangiocarcinoma between 2010 and 2020. The mean age was 64.5 years (±11.8 years) and 44.7% were female. The prevalence of cholangiocarcinoma-specific risk factors was relatively low, including viral hepatitis (15.8%), alcohol abuse (4.4%), and primary sclerosing cholangitis (0.9%). Other risk factors reflected the general population, including obesity (22.1%) and diabetes (23.7%). The majority of tumors were intrahepatic (53.5%) with the remainder extrahepatic distal cholangiocarcinomas (31.6%) and hilar (14.9%). Biliary obstruction (total bilirubin > 3) was seen in 40.4% of patients at presentation. Histologic grade was most commonly moderately (45.6%) or poorly differentiated (46.5%). Mean tumor size was 4.6cm (±2.5cm), and a minority were classified as multifocal (14.0%) or bilobar (3.5%). Patient demographics and tumor characteristics of all patients enrolled in this study are summarized in Table 1. AJCC 8th edition T-stage and N-stage, along with pathological stages are also summarized in Table 1. Two patients (1.9%) were designated as pathological stage IV based upon findings of distant disease on final pathology.

In this study, 36 patients (31.6%) with cholangiocarcinoma underwent TGP. Patients who underwent genetic sequencing were significantly younger than those patients who did not undergo tumor sequencing (60.7 years versus 66.3 years, *p* = 0.02). We did not identify any other differences in demographics or tumor characteristics between sequenced and unsequenced patients. Table 2 summarizes treatment characteristics for all patients in this study. Three-quarters of patients received systemic chemotherapy at some point in their treatment course (72.8%). Radiation therapy was administered to 38.4% of patients in either the neoadjuvant or adjuvant setting. The most common operation was liver resection without bile duct resection (53.5%), and 75.4% of patients were found to have microscopically negative margins (i.e., R0). There were no significant differences in treatment characteristics between those patients who underwent TGP and the remainder of patients. Though most (91%) genetic sequence tests were performed during the mid- (2014–2016) to late-period (2017–2020) of the study, this was not significant (*p* = 0.06).

Table 3 summarizes the TGP results performed on the tumors of the 36 sequenced patients in this study. The breadth of genes sequenced ranged from small gene panels (e.g., 48 genes) to whole exome sequencing (performed in 12 patients). The tumor mutational burden ranged from less than one to five mutations per megabase pair. A total of 41 unique mutations were identified. A median of two actionable mutations was identified in each sample (range 0–9). Targeted sequencing and whole exome sequencing identified a similar number of mutations (mean 2.9 and 1.8, respectively). Common mutations included 12 *TP53* mutations/amplifications (33.3%), 11 *KRAS* mutations (30.6%), five *IDH1/2* mutations (13.9%), and three *BRAF* V600E mutations (8.3%). Three patients had *FGFR* alterations (8.3%). Figure 1 displays the gene plot of identified mutations across 36 sequenced tumors. An average of seven clinical trials per patient were identified (range 0–36). A median of one drug per patient was associated with identified mutations (range 0–11). Among 26 eligible patients (i.e., underwent TGP and had an identified target), six received drugs targeting identified mutations in the setting of recurrent or persistent disease (23%, or 4.4% of the entire cohort).

Patients were followed for a median of 22 months (Interquartile range 12–42). At the last follow-up, 42 patients exhibited no evidence of disease (36.8%), while 38 were alive with disease (33.3%), and 34 had died (29.8%). Median overall survival was estimated as 59 months. Patients who underwent genetic sequence testing had a median overall survival of 42 months, while median survival was not reached among patients without TGP (i.e., greater than 50% surviving at last follow-up) [21]. There was no significant difference in the estimates of median survival between groups (*p* = 0.16). Figure 2 demonstrates the Kaplan Meier curves for survival among patients in this series, including those who underwent TGP and those who did not undergo TGP.

## 4. Discussion

In this single center case series of resected cholangiocarcinoma over the last decade, we found that a third of patients underwent TGP. Testing was more common among younger patients, which may reflect the trend toward use of TGP in patients who have a poor prognosis yet for whom clinicians are searching for a chance at effective therapy. TGP was also more common during the later years of the 2010s, which reflects the increasing use over time as this technology becomes more widely known and available. Among eligible patients, a quarter were treated with targeted therapies related to mutations identified via TGP.

The TGP reported in this study varied in the extent of genome that was sequenced. Commonly available early-generation tests include panels of 48–50 commonly mutated genes. By comparison, FoundationOne testing (Foundation Medicine, Cambridge, MA, USA) has been commercially available since 2012 and provides a panel of several hundred genes. A third of patients in this study were tested with a FoundationOne assay. Lastly, twelve patients in this study were tested with the GEM ExTra whole exome sequence test (Ashion Analytics LLC, Phoenix, AZ, USA). This assay is designed to detect tumor-specific mutations in both DNA and RNA and, therefore, samples variants not only in the genome of cells within the tumor but also the transcriptome. This assay was provided free-of-charge to patients in this study. In the MOSCATO-01 trial, whole exome sequencing identified additional variants in relevant genes in 38% of patients beyond those identified through targeted gene sequencing [19,22]. In this study, both targeted gene sequencing and whole exome sequencing identified a similar mean number of targetable mutations. It is, nonetheless, expected that whole exome sequencing will play an important role in personalized medicine for a wide variety of solid malignancies.

In this study, a median of one drug per patient was associated with efficacy in identified mutations. However, we found that only a quarter of patients who were eligible, were treated with a targeted therapy. This finding might be explained by a number of possible observations. The first is that, while TGP may have become more commonplace, the general knowledge and perceived utility of targeted agents may still be low among medical oncologists. Second, targeted agents are expensive and are likely not covered by many insurers. This may be a primary reason that these targeted agents used in an “off label” fashion are only accessible to a fraction of patients with associated mutations. The third issue raises the question of timing of administration of targeted therapies. In our practice, we find that the use of TGP in patients with resected cholangiocarcinoma most commonly comes into play in the setting of recurrence. This observation bears out in the assessment of comparative survival between sequenced an unsequenced patients. While no significant difference was found, Figure 2 suggests that those patients who are sequenced have the worst survival. While many patients who develop recurrence following resection will be treated with palliative measures, it remains to be seen whether targeted therapies will be integrated into the first-line in the adjuvant setting. It is not known how mutations vary between principally resected and recurrent tumors. Assays such as circulating tumor DNA for detecting molecular residual disease following resection of cholangiocarcinoma may aid in identifying patients that may benefit from targeted therapies prior to clinically evident recurrence.

The most common mutations were *TP53, KRAS, IDH, BRAF,* and *PIK3CA.* Isocitrate dehydrogenase mutations (*IDH1/2*) are common in intrahepatic cholangiocarcinoma (13%) [23,24]. Mutant IDH blocks liver progenitor cells from undergoing hepatocyte differentiation. In vitro dasantanib treated IDH mutated xenografts demonstrate pronounced apoptosis and tumor regression [25]. In a recent phase III clinical trial for patients with progression on chemotherapy, ivosidinib was associated with significant improvement in progression-free survival compared with placebo (ClarIDHy trial, 2.7 months vs. 1.4 months, HR 0.37, 95% CI 0.25–0.54, *p* < 0.0001) [23]. One patient in this study with recurrence of previously resected cholangiocarcinoma was found to have an *IDH1* R132L mutation and is under treatment with ivosidinib.

Fibroblast growth factor receptors (*FGFR*) are a family of transmembrane proteins with tyrosine kinase domains, which are commonly mutated or fused in urothelial, breast, and gynecologic cancers, as well as cholangiocarcinomas (commonly *FGFR2*) [12]. In addition to non-selective tyrosine kinase inhibitors, which have shown activity in cholangiocarcinoma with *FGFR2* fusion, a number of phase II trials have demonstrated efficacy for targeted *FGFR* inhibitors [12]. Selective *FGFR* kinase inhibitor Infigratinib (BGJ398) showed efficacy in some patients with *FGFR2* mutations [26,27]. Among patients receiving Pemigatinib (*FGFR* inhibitor) in the setting of an *FGFR* alteration in cholangiocarcinoma, 35.5% achieved an objective response [28]. Futibatinib (TAS-120, highly selective inhibition of *FGFR1-4*) demonstrated a disease control rate of 79% [29]. Other *FGFR*-targeting agents have demonstrated lesser effectiveness [12].

In this study, three patients undergoing TGP were found to have fibroblast growth factor-related mutations, but none were treated with associated drugs.

*BRAF* mutations are present in 5% of cholangiocarcinomas [30]. These patients have a higher stage of tumor at resection, greater likelihood of lymph node involvement, and worse overall survival compared with non-*BRAF* mutated cholangiocarcinoma. In a basket trial of patients with V600E mutated cancers, *BRAF* inhibitor (dabrafenib) and *MEK* inhibitor (trametinib) combination demonstrated a clinical response in 51% of patients [31]. Two of three patients with V600E mutations in this study were treated with *BRAF/MEK* inhibition.

There are reports of patient response to mTOR inhibition (everolimus) in tumors with *PIK3CA* mutations. This mutation is found in as many as 8% of cholangiocarcinomas and occurred in 8% of our cohort [32]. None, however, were treated with targeted therapies. Other areas of investigation for targeted therapies in cholangiocarcinoma include human epidermal growth factor receptor 2 (*HER2*) and *RNF2* mutations [12].

While there is great optimism about the future of precision medicine for difficult to treat malignancies, there are a number of limitations to this tool as it pertains to cholangiocarcinoma. Some have argued that the genetic heterogeneity of biliary tract cancers may hamper efforts to elucidate which mutational targets will yield effects for patients [33]. Further, there is potential heterogeneity in the frequency of targetable mutations between intra and extrahepatic lesions; however, this study was not intended to detect such a difference. Lack of a “stereotyped” genetic signature to cholangiocarcinoma may slow development of trials of targeted therapies, further compounded by the relative rarity of this tumor overall. While *IDH* mutations and *FGFR2* fusions have promise for targeted therapies, these represent a small minority of identified mutations [12]. An additional limitation of the use of TGP in cholangiocarcinoma is tissue availability [34]. For patients with a locally advanced disease that is unresectable, targeted therapies might provide a chance at disease control but require additional testing after routine pathologic diagnosis. For patients whose diagnosis is made based upon cytologic specimens such as fine needle aspirate or duct brushings, obtaining cell block material or dedicated fine needle biopsy material reserved for genomic or molecular sequence testing is key. Appropriate stewardship of diagnostic materials will require pre-testing coordination to preclude additional or repeated intervention. Lastly, while targeted genomic testing may cost only a few hundred US dollars, whole exome sequencing often costs 2–5 times more per assay. While this limitation may be prohibitive at present, we take the approach that whole exome-based TGP may provide valuable information for patients about emerging targets and new clinical trials on the immediate horizon.

## 5. Conclusions

Overall, our study shows that the use of TGP in cholangiocarcinoma has increased over the past decade. While the majority of patients were found to have potentially actionable mutations, this testing resulted in an altered treatment in only a quarter of eligible patients. The findings of this study must be interpreted in the context of a single institution experience. While our care teams had ready access to whole exome sequencing for several years before it was commonly available elsewhere, the application of TGP and choice to prescribe targeted therapies can be expected to vary between providers and institutions. This sample of patients may not represent the natural distribution of mutations seen in the larger population of patients with cholangiocarcinoma. It should also be emphasized that tumors from surgically resected patients in this study may not have the same mutational burden seen in locally advanced or metastatic disease. Furthermore, some patients may have been treated with targeted therapies at other institutions following resection at our quaternary center and, therefore, may not have been captured by this analysis.

Future directions for targeted therapies in cholangiocarcinoma include the continued trial of drugs among patients with unique mutations. To address the issue of limited tissue available for TGP, ‘liquid biopsies’ will likely play a role in identifying tumor-based mutations through circulating tumor DNA [35]. While patients await more effective first-line agents for cholangiocarcinoma, precision methods may play a more active role in up-front therapy for patients with resected tumors [12]. To facilitate this, payors and providers will need to continue the trend of increasing application of TGP in the coming years.

## Figures and Tables

**Figure 1 cancers-13-04062-f001:**
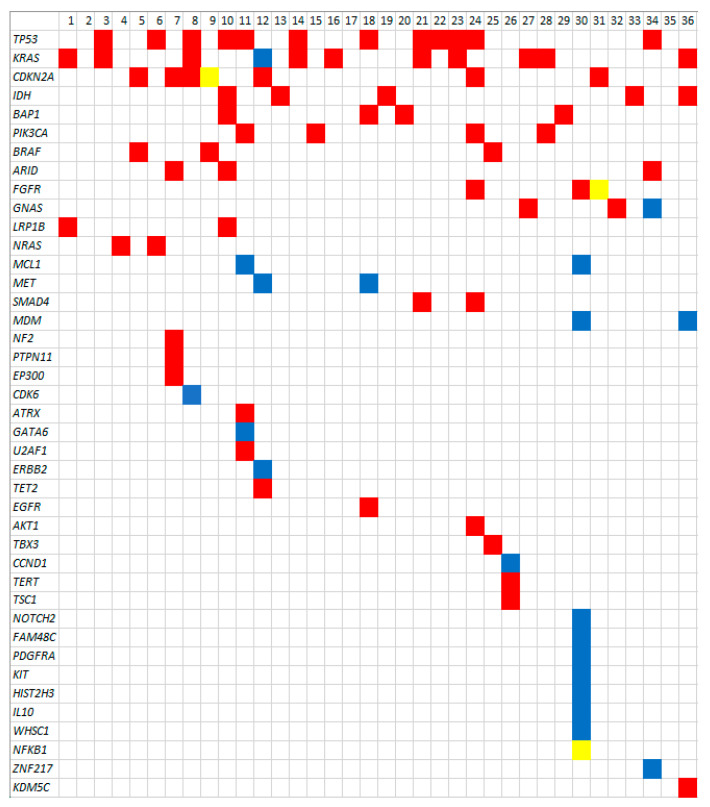
Gene plot of all identified mutations among 36 sequenced cholangiocarcinomas. Red squares represent gene mutations, blue squares represent gene fusion, and yellow squares represent gene loss.

**Figure 2 cancers-13-04062-f002:**
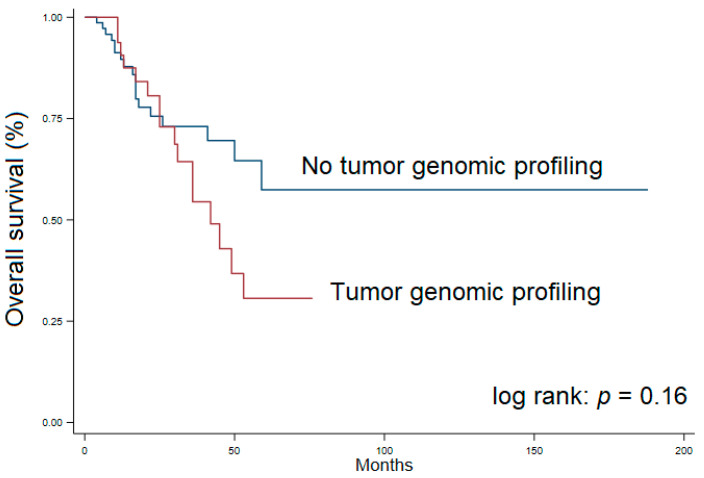
Kaplan Meier curves for overall survival among patients who underwent tumor genomic profiling (red line) and did not undergo tumor genomic profiling (blue line).

**Table 1 cancers-13-04062-t001:** Demographics, distribution of cholangiocarcinoma risk factors, clinical characteristics, and pathologic stage of patients with resected cholangiocarcinoma, including those who did and did not receive genomic sequencing of their tumor.

	All Patients *n* = 114	Patients Receiving Sequencing *n* = 36	Patients Not Receiving Sequencing *n* = 78	*p*-Value
Demographics				
Mean age (years ± SD) (at diagnosis)	64.5 ± 11.8	60.7 (12.9)	66.3 (10.9)	0.02
Female, *n* (%)	51 (44.7)	17 (47.2)	34. (43.6)	0.72
Race, *n* (%)				
Asian	30 (26.3)	9 (25.0)	21 (26.9)	0.71
Black	0 (0)	0 (0)	0 (0)	
Hispanic	30 (26.3)	8 (22.2)	22 (28.2)	
Middle Eastern	0 (0)	0 (0)	0 (0)	
White	54 (47.4)	19 (52.3)	35 (44.9)	
Cholangiocarcinoma risk factors, *n* (%)				
Viral hepatitis	18 (15.8)	5 (13.9)	13 (16.7)	0.71
Primary sclerosing cholangitis	1 (0.9)	1 (2.8)	0 (0)	0.14
Alcohol abuse	5 (4.4)	2 (5.6)	3 (3.9)	0.68
Obese (BMI > 30)	25 (22.1)	6 (16.7)	19 (24.7)	0.34
Diabetes	27 (23.7)	5 (13.9)	22 (28.2)	0.10
Clinical characteristics, *n* (%)				
Obstructive jaundice at diagnosis	46 (40.4)	13 (36.1)	33 (42.3)	0.53
Tumor site				0.69
Intrahepatic, *n* (%)	61 (53.5)	21 (58.3)	40 (51.3)	
Hilar, *n* (%)	17 (14.9)	4 (11.1)	13 (16.7)	
Extrahepatic distal, *n* (%)	36 (31.6)	11 (30.6)	25 (32.1)	
Tumor grade, *n* (%)				0.73
Well differentiated	9 (7.9)	2 (5.6)	7 (9.0)	
Moderately differentiated	52 (45.6)	18 (50.0)	34 (43.6)	
Poorly differentiated	53 (46.5)	16 (44.4)	37 (47.4)	
Mean tumor size, cm (SD)	4.6 (2.5)	4.5 (2.9)	4.4 (2.3)	0.80
Multifocal disease	16 (14.0)	8 (22.2)	8 (10.3)	0.29
Bilobar disease	4 (3.5)	1 (2.8)	3 (3.9)	0.77
T-stage				0.40
1	26 (22.9)	8 (22.8)	18 (24.3)	
2	40 (36.7)	16 (45.7)	24 (32.4)	
3	40 (36.7)	11 (31.4)	29 (39.2)	
4	3 (2.8)	0 (0)	3 (4.1)	
N-stage				0.34
0	57 (58.2)	22 (68.8)	35 (53.0)	
1	37 (37.8)	9 (28.1)	28 (42.4)	
2	4 (4.1)	1 (3.1)	3 (4.6)	
AJCC 8th Edition—Pathological Stage				0.40
IA	12 (11.1)	3 (8.8)	9 (12.2)	
IB	28 (25.9)	7 (20.6)	21 (28.4)	
IIA	25 (23.2)	12 (35.3)	4 (5.4)	
IIB	28 (25.9)	7 (20.6)	21 (28.4)	
IIIA	10 (9.3)	2 (5.9)	8 (10.8)	
IIIB	21 (19.4)	7 (20.6)	14 (18.9)	
IIIC	3 (2.8)	0 (0)	3 (4.1)	
IV	2 (1.9)	0 (0)	2 (2.7)	

**Table 2 cancers-13-04062-t002:** Therapies and follow-up for patients with resected cholangiocarcinoma.

	All Patients *n* = 114	Patients Receiving Sequencing *n* = 36	Patients Not Receiving Sequencing *n* = 78	*p*-Value
Systemic chemotherapy	83 (72.8)	28 (77.8)	55 (70.5)	0.42
Radiation therapy	43 (38.4)	15 (41.7)	28 (36.8)	0.62
Surgery				0.64
Liver resection without bile duct resection	61 (53.5)	21 (58.3)	40 (51.3)	
Resection requiring bile duct resection/reconstruction	21 (18.4)	7 (19.4)	14 (18.0)	
Pancreaticoduodenectomy	32 (28.1)	8 (22.2)	24 (30.8)	
Margin status				0.32
R0	86 (75.4)	26 (72.2)	60 (76.9)	
R1	25 (21.9)	10 (27.8)	15 (19.2)	
R2	3 (2.6)	0 (0)	3 (3.9)	
Period of treatment				0.06
Early (2010–2013)	25 (21.9)	3 (8.3)	22 (28.2)	
Mid (2014–2016)	41 (36.0)	16 (44.4)	25 (32.1)	
Late (2017–2020)	48 (42.1)	17 (47.2)	31 (39.7)	
Median overall survival—months (25%, 75%)	59 (25, NR)	42 (25, NR)	NR (26, NR)	0.16

**Table 3 cancers-13-04062-t003:** Summary of patients undergoing molecular pathological assessment, identified mutations, associated drugs and trials, therapy received, and patient survival.

	Genes Assessed (WE: Whole Exome)	Actionable Mutations Identified	Tumor Mutational Burden (per MBP)	Identified Mutations/Alterations	Drugs Associated with Identified Mutations	Trials Identified	Therapy Received	Survival (Months) * AWD, † NED at Last Follow-up
*1.*	324	2	-	*KRAS* G12S *LRP1B* D478N	*KRAS* Cobimetinib Trametinib	8	Gemcitabine/Cisplatinum 5-FU FOLFOX Pembrolizumab	53
*2.*	48	0	-	-	0	0	Gemcitabine/Cisplatinum	68
*3.*	324	2	-	*KRAS* G12V *TP53* R248Q	*KRAS* *Trametinib*	6	Gemcitabine/Cisplatinum Capecitabine	21
*4.*	WE	1	1	*NRAS* Q61R	0	11	Gemcitabine/Cisplatinum FOLFOX	76 *
*5.*	324	2	-	*BRAF* V600E *CDKN2A/B*	*BRAF* Dabrafenib Regorafenib Trametinib Vemurafenib	10	Gemcitabine/Cisplatinum Capecitabine Dabrafenib/Trametinib	31
*6.*	50	2	-	*TP53* R213 *NRAS* G12D	*MEK* Trametinib *mTOR* Everolimus Temsirolimus	5	Gemcitabine/Cisplatinum	13
*7.*	324	6	-	*ARID1A* E1763 & Q372 *NF2* 447 *PTPN11* G503V *CDKN2A* *EP300*	*NF2* Everolimus Lapatinib Temsirolimus Trametinib *PTPN11* Trametinib	10	Sorafenib Pembrolizumab	45
*8.*	WE	4	1	*KRAS* G12V *TP53* S125G *CDK6* amplification *CDKN2A* L16	*CDK6* Abemaciclib Palbociclib Ribociclib	28	-	68 *
*9.*	324	2	-	*BRAF* V600E CDKN2A loss	*BRAF* Cobimetinib Dabrafenib Regorafenib Trametinib Vemurafenib	4	Gemcitabine/Cisplatinum	59 *
*10.*	324	5	-	*ARID1A* Y1719 *IDH2* R172W *TP53* Y220C *BAP1* 123-1 *LRP1B* R295	None	1	-	8 *
*11.*	324	6	-	*PIK3CA* M1004I *TP53* C141W *ATRX* A419V *GATA6* amplification *MCL1* amplification *U2AF1* S34F	*PIK3CA* Everolimus Temsirolimus	4	Gemcitabine/Cisplatinum Pembrolizumab	53 *
*12.*	324	7	5	*ERBB2* amplification *KRAS* amplification *MET* amplification *CDKN2A* p16INK4a & p14ARF *TET2* R1572W	*ERBB2* Afatinib Lapatinib Neratinib Pertuzumab Traztuzumab Ado-traztuzumab Traztuzumab-dkst *KRAS* Cobimetinib Trametinib *MET* Cabozantinib Crizotinib	19	Gemcitabine/Cisplatinum FOFOX Capecitabine	35 *
*13.*	324	1	-	*IDH1*	None	0	Gemcitabine/Cisplatinum CAPOX	36
*14.*	-	2	-	*KRAS* *TP53*	None	0	Gemcitabine/Cisplatinum FOLFOX Atezolizumab Cobimetinib	42
*15.*	49	1	-	*PIK3CA* E542K	None	0	-	10 *
*16.*	48	1	-	*KRAS* G12V	None	0	Gemcitabine/Cisplatinum FOLFIRI Capecitabine	25
*17.*	WE	0	<1	-	None	0	Gemcitabine/Cisplatinum FOLFOX 5-FU	41 *
*18.*	324 & 48	5	3	*EGFR* A289V *BAP1* V616 *TP53* V173M *MET* amplification	*MET* Crizotinib Cabozantinib *EGFR* Pantitumumab Osimertinib Lapatinib Gefitinib Erlotinib Cetuximab Afatinib	10	Gemcitabine/Cisplatinum FOLFOX FOLFIRI	11
*19.*	159	1	5	*IDH1* R132L	RRM1 positive Gemcitabine	4	Gemcitabine/Cisplatinum Capecitabine	43 *
*20.*	159	1	5	*BAP1* E278	None	0	Gemcitabine/Paclitaxel FOLFOX FOLFIRI Sorafenib	30
*21.*	48	3	-	*KRAS* G12D *TP53* H193R *SMAD4* R361C	None	0	-	11
*22.*	WE	1	<1	*TP53* V216G	None	1	Gemcitabine/Cisplatinum	33 *
*23.*	324	3	-	*KRAS* G12V *TP53* A276D *KDM6A* P334	*KRAS* Cobimetinib Trametinib	0	-	18 *
*24.*	324	9	4	*PIK3CA* E545K *AKT1* W80R *FGFR2* C382R *CDKN2A/B* loss *SMAD4* R361C & R361H *TP53* R213	*AKT1* & *PIK3CA* Everolimus Temsirolimus *FGFR2* Pazopanib Ponatinib	19	Gemcitabine/Cisplatinum Capecitabine	12
*25.*	324	2	-	*BRAF* V600E *TBX3* 942-1G	*BRAF* Cobimetinib Dabrafenib Trametinib Regorafenib Vemurafenib	9	Gemcitabine Capecitabine Pembrolizumab Dabrafenib/Trametinib Lenvatinib	25
*26.*	WE	3	2	*CCND1* amplification *TERT* C124C4 *TSC1* Y48	*CCND1* Abemaciclib Palbociclib Ribociclib *TSC1* Everolimus Temsirolimus	17	Pembrolizumab	14 *
*27.*	49	2	-	*KRAS* G13D *GNAS* R201H	None	36	Gemcitabine/Cisplatinum Capecitabine CAPOX	25 *
*28.*	WE	2	1	*KRAS* G12D *PIK3CA* H1047R	*PIK3CA* Copanlisib Everolimus Temsirolimus	10	Gemcitabine/Cisplatinum Capecitabine	22 *
*29.*	WE	1	<1	*BAP1* F15_T16	0	4	Gemcitabine/Cisplatinum	17
*30.*	168	0	0	*MDM4* amplification *NOTCH2* amplification *FAM48C* amplification *PDGFRA* amplification *KIT* amplification *HIST2H3D* amplification *HIST2H3C* amplification *MCL1* amplification *IL10* amplification *FGFR3* amplification *WHSC1* gain *FGFR2* loss *NFKB1A* loss *FGFR3* TACC3	0	0	Gemcitabine Floxuridine	49
*31.*	WE	3	1	*CDKN2A* H83N *FGF3* amplification *FGF4* amplification	*FGF3* Sorafenib *FGF4* Pazopanib Sorafenib	10	Gemcitabine/Cisplatinum Capecitabine	15 ^†^
*32.*	WE	1	<1	*GNAS* R844C	None	6	-	7 ^†^
*33.*	WE	1	1	*IDH1* R132C	*IDH1* Ivosidinib	0	Capecitabine	3 *
*34.*	48	4	-	*TP53* P152 *ARID2* F1537 *GNAS* amplification *ZNF217* amplificaiton	None	6	Gemcitabine/Cisplatinum 5-FU Pembrolizumab	36
*35.*	WE	0	<1	-	None	0	Gemcitabine/Cisplatinum	22 ^†^
*36.*	WE	4	<1	*KRAS* G12D *IDH1* R132L *KDM5C* G1452 *MDM2* amplification	*IDH1* Ivosidinib *KDM5C* Sunitinib	15	Gemcitabine/Cisplatinum Ivosidinib	26 *

WE: Whole exome. The * stands for “alive with disease” and the † stands for “No evidence of disease”.

## Data Availability

Data is available on request due to restrictions regarding patient privacy.

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
