# Peer review of "Patterns of Whole Exome Sequencing in Resected Cholangiocarcinoma"

_cancers, 2021, doi:10.3390/cancers13164062_

Round 1
Reviewer 1 Report
The authors analyzed whole exome sequencing in resected cholangiocacinoma in a retrospective cohort from 2010 to 2020.
Comments
- The surgical policy of cholangiocarcinoma should be declared. How many patients with cholangiocarcinoma were unresectable? Do any patient receive neoadjuvant therapy before surgery and included for exome analysis?
- Do all patients with resected cholangiocarcinoma offer the sequencing service? What factors determine the service to be performed or not? insurance coverage or trial?
- A separate section of statistical analysis is needed.
- Page 7, Line 120, P= 0.06 in this small cohort suggest a borderline significance due to insufficient power limited by small case number, probably "not significant". What is the rationale to decide the two cut points of the period?
- In Table 3, what do WE mean?
- Can you show us the survival curves? Is the sequencing result stratify the curve differently?
Author Response
- The surgical policy of cholangiocarcinoma should be declared. How many patients with cholangiocarcinoma were unresectable? Do any patient receive neoadjuvant therapy before surgery and included for exome analysis?
Thank you for this comment. This analysis did not include patients who were considered unresectable or metastatic at diagnosis. Those patients who are resectable represent a minority of cases. Many patients with cholangiocarcinoma are treated with neoadjuvant chemotherapy at our institution however we did not assess the temporal relationship of chemotherapy and surgery in this series.
- Do all patients with resected cholangiocarcinoma offer the sequencing service? What factors determine the service to be performed or not? insurance coverage or trial?
This study was designed to demonstrate the usual practice of clinicians within our system over the past decade. It appears that even at present, that some patients are not offered tumor genomic profiling although this has certainly increased over the last year. We found that patients who were offered TGP were on average somewhat younger. We have also observed that TGP is commonly employed in the setting of recurrence. In this study, however, we did not specifically analyze the time point in treatment that TGP was employed. There are certainly many factors in adoption of TGP including insurance and provider familiarity. In the final years of this study, whole exome sequencing was available free of charge to most patients with solid organ malignancies through clinical trial enrollment at our institution.
- A separate section of statistical analysis is needed.
The statistical analysis reported in this study is described in lines 88-90.
- Page 7, Line 120, P= 0.06 in this small cohort suggest a borderline significance due to insufficient power limited by small case number, probably "not significant". What is the rationale to decide the two cut points of the period?
The choice to measure use of TGP in thirds of the decade was made a priori. When the use of TGP is measured in two eras (early and late), the increase in use over the decade was not significant (p = 0.12)
- In Table 3, what do WE mean?
WE is an abbreviation for whole exome. This has been defined in the revised submission.
- Can you show us the survival curves? Is the sequencing result stratify the curve differently?
Per the request, we have included a survival curve in the revised manuscript, stratified by use of TGP. This is now listed as figure 2 and is described in the revised manuscript. These curves demonstrate that TGP is being performed upon those patients with the worst survival. This is addressed in the revised manuscript.
Reviewer 2 Report
The authors reported results of the tumor genetic profiling (TGP) of cholangiocarcinoma. Thirty-six patients undergone TGP showed a mean of 3.1 actionable mutations per patient. Mutations aligned with a median of 1 drug per patient. Common mutations included TP53 (33%), KRAS (31%), IDH1/2 (14%), FGFR (14%), and BRAF (8%). Targeted therapies were administered in only 4% of patients (23% of eligible sequenced patients). After a median 22 months, 23% had recurrence and 29% were deceased. The authors concluded that TGP for cholangiocarcinoma has increased over the last decade with targeted therapies identified in most sequenced tumors; impacting treatment in a quarter of eligible patients. Precision medicine will play a central role in the future care of cholangiocarcinoma.
This study was mostly in a descriptive nature for their clinical experience of TGP of cholangiocarcinoma. Apparently small number of genetically examined cases may hamper the significance of this study, however, because of rarity of cholangiocarcinoma, it seems to be worth reporting as a real-world data for clinical sequencing of cholangiocarcinoma. There are some minor concerns.
- How were the actionable mutations determined?
- In Table 3, there are some typos in mutation calls. What does “CDKN2 A/B” mean in case 4? What were mutations “TP53 R213” in case 6; “ARID1A E1763 & Q372” in case 7; “CDKN2A L16” in case 8; BAP1 123-1 and LRP1B R295 in case 10; IDH1 in case 13; KRAS, TP53 in case 14, etc.?
- In page 13, line 142, regarding the description “Patients who underwent genetic sequence testing had a median overall survival of 42 months while median survival was not reached among patients without TGP”, why was the median survival of patients without TGP not disclosed?
- In page 16, line 167, the description “In this study, both targeted gene sequencing and whole exome sequencing identified a similar mean number of targetable mutations (2.9 and 1.8, respectively)” should be moved to the results section.
- In page 16, line 171, the description “However, we found that only 23% of eligible patients who underwent TGP were treated with a drug targeting tumor-specific mutations in this series (4.4% of the entire cohort)” should be moved to the results section.
- Were there any association between specific gene mutations and clinicopathological features including survivals?
Author Response
- How were the actionable mutations determined?
Thank you for this question. Our research team did not independently assess actionability of mutations. The definition of clinically significant or actionable mutations was defined by each individual assay. This topic is relevant as the number of mutations known to be clinically significant will increase over time. Actionable mutations are those whereby there are targetable agents.
- In Table 3, there are some typos in mutation calls. What does “CDKN2 A/B” mean in case 4? What were mutations “TP53 R213” in case 6; “ARID1A E1763 & Q372” in case 7; “CDKN2A L16” in case 8; BAP1 123-1 and LRP1B R295 in case 10; IDH1 in case 13; KRAS, TP53 in case 14, etc.?
The mutations reported in this study are described according to the level of detail provided in the tumor genomic profiling report. Because genomic testing was performed by a variety of assays, the convention of mutation reporting varied somewhat from patient to patient. However, the general convention is the name of the altered/mutated gene (in italics) followed by the position of mutation. Thus the TP53 R213 mutation was identified in the TP53 gene at the R213 position.
- In page 13, line 142, regarding the description “Patients who underwent genetic sequence testing had a median overall survival of 42 months while median survival was not reached among patients without TGP”, why was the median survival of patients without TGP not disclosed?
As a point of clarification, at the time of data collection, for the group of patients who did not undergo TGP, more than 50% were still alive at last follow up, therefore median survival is not defined (i.e. not reached). This has been clarified in the revised manuscript.
- In page 16, line 167, the description “In this study, both targeted gene sequencing and whole exome sequencing identified a similar mean number of targetable mutations (2.9 and 1.8, respectively)” should be moved to the results section.
Thank your for this comment. This has been moved to the results section in the revised manuscript.
- In page 16, line 171, the description “However, we found that only 23% of eligible patients who underwent TGP were treated with a drug targeting tumor-specific mutations in this series (4.4% of the entire cohort)” should be moved to the results section.
This has been moved to the results section.
- Were there any association between specific gene mutations and clinicopathological features including survivals?
Please see our response above, the objective of this study was to describe the use of TGP in resected cholangiocarcinoma and to assess its impact upon the treatment choices for these patients. We did not intend to, nor did we design the study around detecting differences in survival related to particular mutations or targeted therapies. It was beyond the scope of this study to assess clinicopathologic characteristics of resected specimens. Further, as each unique mutation occurred in at most 12 patients, identifying associations within this cohort would be unlikely.
Reviewer 3 Report
In this paper, the authors conducted a retrospective study on 114 cholangiocarcinoma patients, of which 36 underwent sequencing, to identify the frequency of therapeutically targetable mutations and verify in which cases target therapies were actually used.
A median of 2 actionable mutations were identified per patient, with a prevalence of TP53, KRAS, IDH1/2 and BRAF mutations. Among 26 eligible patients, only 6 received therapies targeting identified mutations, but mainly in the context of recurrent or persistent disease, not as first-line treatment. This emphasizes the little use that is still made today of tumor genetic profiling and of target therapies in general.
The study is well conducted and quite interesting. Despite this, there are some unresolved questions:
- Even if the cohort of examined patients is very small, is it possible to notice a difference in the frequency or type of mutations depending on the clinical-pathological characteristics of patients?
-According to the authors, is there any utility in performing whole exome sequencing compared to targeted gene sequencing? Despite the obvious interest from a scientific research point of view, if we identify mutations for which to date there are no available target therapies, what usefulness can whole exome sequencing have in clinical practice?
-Which are the costs of these 2 types of sequencing (whole exome vs targeted gene)? Is it possible to apply them in daily clinical practice? The author should also discuss this point.
- It would be interesting to enlarge the cohort of examined patients, perhaps also drawing on the sequencing information available in public databases. Has this point been taken into consideration?
Author Response
Even if the cohort of examined patients is very small, is it possible to notice a difference in the frequency or type of mutations depending on the clinical-pathological characteristics of patients?
The objective of this study was to describe the use of TGP in resected cholangiocarcinoma and to assess its impact upon the treatment choices for these patients. We did not intend to, nor did we design the study around detecting differences in survival related to particular mutations or targeted therapies. It was beyond the scope of this study to assess clinicopathologic characteristics of resected specimens. Further, as each unique mutation occurred in at most 12 patients, identifying associations within this cohort would be unlikely. While this would be very interesting to know and could potentially reveal important features of these unique mutations. However, we found that pathology reports are not codified and therefore commonly do not specify the clinicopathologic subtype of cholangiocarcinoma. Further investigation in this area is warranted as it may be an avenue of discovery.
According to the authors, is there any utility in performing whole exome sequencing compared to targeted gene sequencing? Despite the obvious interest from a scientific research point of view, if we identify mutations for which to date there are no available target therapies, what usefulness can whole exome sequencing have in clinical practice?
While we provide no evidence in this study that whole exome sequencing is superior to more limited assessment of tumor mutation, we take an optimistic point of view that more information for the patient can be of greater use in the journey to identify an effective treatment for this difficult to treat disease. This includes access to ongoing clinical trials as well as alignment with agents that may be developed in the future within a short time horizon.
-Which are the costs of these 2 types of sequencing (whole exome vs targeted gene)? Is it possible to apply them in daily clinical practice? The author should also discuss this point.
We find that most insurance carriers will cover targeted TGP. These tests cost a few hundred dollars and are a feasible part of daily clinical practice for surgeons and medical oncologists, alike. Whole exome sequencing is approximately 2-5x more expensive but was uniquely available to many patients at our institution. We cannot estimate the frequency of insurance coverage for this type of testing but expect that it will become more commonplace as more targets are identified. This point has been clarified in the revised manuscript.
It would be interesting to enlarge the cohort of examined patients, perhaps also drawing on the sequencing information available in public databases. Has this point been taken into consideration?
We agree that large database analyses of tumor mutation in cholangiocarcinoma would be incredibly fruitful for discovering important associations with regard to survival and treatment outcomes, however to our knowledge these mutations are not reported in most registries.
Round 2
Reviewer 1 Report
I have no other comments.